# A phase I dose-finding design with incorporation of historical information and adaptive shrinking boundaries

Chen Li[1], Haitao Pan[2]*

1 Department of Health Statistics, School of Preventive Medicine, Fourth Military Medical University, Xi'an, Shaanxi, China, 2 Department of Biostatistics, St. Jude Children's Research Hospital, Memphis, TN, United States of America

* haitao.pan@stjude.org

**Data Availability Statement:** All relevant data are within the paper and supporting information file.

## Abstract

Although many novel phase I designs have been developed in recent years, few studies have discussed how to incorporate external information into dose-finding designs. In this paper, we first propose a new method for developing a phase I design, Bayesian optimal interval design (BOIN)[Liu S et al. (2015), Yuan Y et al. (2016)], for formally incorporating historical information. An algorithm to automatically generate parameters for prior set-up is introduced. Second, we propose a method to relax the fixed boundaries of the BOIN design to be adaptive, such that the accumulative information can be used more appropriately. This modified design is called adaptive BOIN (aBOIN). Simulation studies to examine performances of the aBOIN design in small and large sample sizes revealed comparable performances for the aBOIN and original BOIN designs for small sample sizes. However, aBOIN outperformed BOIN in moderate sample sizes. Simulation results also showed that when historical trials are conducted in settings similar to those for the current trial, their performance can be significantly improved. This approach can be applied directly to pediatric cancer trials, since all phase I trials in children are followed by similar efficient adult trials in the current drug development paradigm. However, when information is weak, operating characteristics are compromised.

## 1 Introduction

In the field of drug development, there is high interest in conducting clinical trials using designs that can enable the incorporation of external information, such as prior or historical information, with trial data to save sample sizes, improve the power, and expedite the trial process. Several studies have focused on developing designs that incorporate external information for phase II or III trials, for example, meta-analytic power prior–based multiple historical sources [1], hierarchical shrinkage method for basket trials [2–4], calibrated power prior for biosimilar trials [5], and Bayesian designs for confirmatory trials [6, 7]. All the above research designs have focused on phase II trials and beyond. Very few studies have discussed how to

**Funding:** The funders had no role in study design, data collection and analysis, decision to publish, or preparation of the manuscript.

**Competing interests:** The authors have declared that no competing interests exist.

incorporate external information for phase I trials, though it is known that phase I trials are crucial because all appropriate evaluations of promising new agents in phase II or III trials have to rely on well-conducted phase I trials. On the other hand, in some settings wherein historical information is available (e.g., pediatric clinical trials), the suggested starting dose is 80% of the dose recommended for adults. Due to ethical constraints and a typically small number of patients in pediatric trials, it is essential to know how to formally incorporate prior knowledge from adult trials. Petit et al. [8] proposed a method to extrapolate pharmacokinetic information from the adult population to the pediatric population in dose-finding trials. Their method focused on phase I/II trials that jointly modeled toxicity and efficacy by using the continual reassessment model (CRM). In contrast to that study, our study considers how to use historical information to inform the prior elicitation for phase I trials only. Our proposed method is based on the Bayesian optimal interval design (BOIN) framework [9, 10].

The BOIN design's escalation/de-escalation decisions are based on two boundaries. Given the DLT target, the two boundaries are fixed (derived by minimizing the overall decision error rate). However, in some situations, we might need to have an unbalanced control of misallocation of patients to under-toxic and over-toxic dose levels. By having accumulative information, we could have a better understanding of the toxicity rate for each dose level tried; fixed boundaries cannot reflect these dynamics. The second goal of this study is to propose flexible boundaries that can change during the trial process. This design is termed as adaptive BOIN (aBOIN).

The rest of the paper gives a brief introduction to the BOIN design, followed by a methodology proposed to incorporate external information based on the BOIN design framework. Next, an approach for extending the BOIN with fixed boundaries to the aBOIN design with non-fixed boundaries is proposed. Empirical findings are shown by comprehensive simulations with derivation of the theoretical properties. The paper ends with a final discussion.

## 2 Brief introduction to the BOIN design

The BOIN design proposed by Liu and Yuan in 2015 [9] is simple to implement and is similar to the 3+3 design, but is much more flexible, and its operating characteristics are superior to those of more complex model-based methods. An R package (BOIN), a stand-alone graphical user interface–based software, and Shiny app (www.trialdesign.org) have been developed, which are freely accessible to users.

The BOIN design can be summarized as follows:

(a). Patients in the first cohort are treated with the lowest or a pre-specified dose level.

(b). Let $\hat{p}_j$ be the observed toxicity rate at the current dose. To assign a dose to the next cohort of patients,

- if $\hat{p}_j \leq \lambda_1$, we escalate the dose level to $j + 1$,

- if $\hat{p}_j \geq \lambda_2$, we de-escalate the dose level to $j - 1$, or

- otherwise, i.e. $\lambda_1 < \hat{p}_j < \lambda_2$, we retain the same dose level, $j$.

  To ensure that dose levels of treatment always remain within the pre-specified dose range, the dose escalation or de-escalation rule needs to be adjusted for the lowest or highest levels of $j$; for example, if $j = 1$ and $\hat{p}_j \geq \lambda_2$ or $j = J$ and $\hat{p}_j \leq \lambda_1$, the dose remains at the same level, $j$.

(c). This process continues until the maximum sample size is reached or the trial is terminated because of excessive toxicity, as described next.

The selection of interval boundaries $\lambda_1$ and $\lambda_2$ is critical, because two parameters essentially determine the operating characteristics of the design. The BOIN design is optimal in the sense that it selects $\lambda_1$ and $\lambda_2$ to minimize incorrect decisions of dose escalation and de-escalation during the trial.

By using $p_j$ to denote the true toxicity probability of dose level $j$ for $j = 1, \ldots, J$, three point hypotheses are formulated:

$$H_{0j} : p_j = \phi,$$

$$H_{1j} : p_j = \phi_1,$$

$$H_{2j} : p_j = \phi_2,$$

where $\phi_1$ denotes the highest toxicity probability that is deemed sub-therapeutic (i.e., below the MTD) such that dose escalation is required, and $\phi_2$ denotes the lowest toxicity probability that is deemed overly toxic, such that dose de-escalation is required.

Under the Bayesian paradigm, each hypothesis was assigned an equal prior probability, denoted as $\pi_{kj} = \mathrm{pr}(H_{kj})$, $k = 0, 1, 2$. The probability of making an incorrect decision (the decision error rate) is minimized when

$$\lambda_1 = \frac{\log\left(\frac{1-\phi_1}{1-\phi}\right)}{\log\left\{\frac{\phi(1-\phi_1)}{\phi_1(1-\phi)}\right\}} \tag{1}$$

$$\lambda_2 = \frac{\log\left(\frac{1-\phi}{1-\phi_2}\right)}{\log\left\{\frac{\phi_2(1-\phi)}{\phi(1-\phi_2)}\right\}}. \tag{2}$$

Details can be found in [9, 10].

## 3 BOIN design with incorporating external information

Viele et al. [11] said, "Clinical trials rarely, if ever, occur in a vacuum. Generally, large amounts of clinical data are available prior to the start of a study". Although the phase I trial is considered the first-in-human study for identifying the MTD, there is still possible information that we can use to enhance our understanding of the toxicity profile for experimented drugs, for example, in the aforementioned phase I pediatric trials or rare diseases occurring in limited patient populations. Another point of view is that well-conducted phase I studies can increase the precision of phase II dose recommendation. High failure rates for late-phase studies can be due to flawed phase I studies. Efficiently using the prior or historical information provides an opportunity to improve the phase I study.

By using the BOIN design framework, prior information can be incorporated naturally via only modifying (1) and (2) by using the following formula:

$$\lambda_1 = \frac{\log\left(\frac{1-\phi_1}{1-\phi}\right) + n_j^{-1}\log\left(\frac{\pi_{1j}}{\pi_{0j}}\right)}{\log\left\{\frac{\phi(1-\phi_1)}{\phi_1(1-\phi)}\right\}} \tag{3}$$

$$\lambda_2 = \frac{\log\left(\frac{1-\phi}{1-\phi_2}\right) + n_j^{-1}\log\left(\frac{\pi_{1j}}{\pi_{0j}}\right)}{\log\left\{\frac{\phi_2(1-\phi)}{\phi(1-\phi_2)}\right\}}. \tag{4}$$

Comparing (3) and (4) to (1) and (2), prior parameters are incorporated in the above formulas by using prior probabilities of the 3-point hypotheses: $\pi_{0j}$, $\pi_{1j}$, and $\pi_{2j}$.

By notations, assuming there are $J$ dose levels, for each dose level $j$, $j = 1, \cdots, J$, we have three prior probability vectors for $\pi_{0,j}$, $\pi_{1,j}$ and $\pi_{2,j}$, $j = 1, \cdots, J$, associated with three point hypotheses $H_{0j}$, $H_{1j}$, $H_{2j}$, $j = 1, \cdots, d$, which are defined in BOIN introduction. All the prior probabilities can be presented explicitly by the following Table 1:

On the basis of data in Table 1, we propose an approach to elicit values for these cells prior to the trial study. If there is strong confidence that dose $D_j$ is closest to the target DLT rate, we assign a larger probability to $\pi_{0,j}$ (e.g., 0.6) and then assign $\pi_{1,j}$, $\pi_{2,j}$ to be equally half of the rest of the probability (e.g., 0.4/2 = 0.2). Here, we believe a priori with 60% confidence that dose $D_j$ would be the MTD and 20% confidence that this dose would be under-dosing or over-dosing. We define $\text{odds}_j$ to be $\frac{\pi_{1,j}}{\pi_{2,j}}$.

By eliciting values of the remaining cells in Table 1, we pre-specify a probability vector for $H_0$, that is, $(\pi_{0,1}, \cdots, \pi_{0,J})$. We emphasize here that pre-specification of the probability vector for $H_0$ is feasible. For example, if we have strong evidence that one dose is near to the MTD, as in pediatric trials, because MTDs in children and adults correlate strongly and 80% of the adult dose is recommended as the starting dose for children, the investigator can effortlessly select with high confidence the dose that can be the MTD and also other doses. If there is weak prior knowledge, equally likely probabilities can be assigned to this vector.

When eliciting values for two probability vectors of $H_1$ and $H_2$, the two vectors need to be in decreasing and increasing orders, respectively. This is because $H_1$ refers to the under-dosing hypothesis; therefore, probabilities of believing in $H_1$ would decrease when dose level increases and vice versa for the probability vector for $H_2$. For example, for a trial with five dose levels, if we assign the probability vector to $H_0$ to be $\pi_{0,1} = 0.05$, $\pi_{0,2} = 0.15$, $\pi_{0,3} = 0.6$, $\pi_{0,4} = 0.15$, $\pi_{0,5} = 0.05$, then $\pi_{0,1} = 0.05$ it means we have very low confidence that the first dose is the MTD. Since the first dose is the lowest dose among the five doses, the $\pi_{1,1}$ should be the highest among $(\pi_{0,1}, \pi_{1,1}, \pi_{2,1})$, since it is the safest dose level; that is, we have high confidence that the first dose will lie in the interval defined by the hypothesis $H_1$, which corresponds to the under-dose interval. As an example, let us assign values $(\pi_{0,1} = 0.05, \pi_{1,1} = 0.85, \pi_{2,1} = 0.10)$ to them by considering the constraint $\pi_{0,j} + \pi_{1,j} + \pi_{2,j} = 1$, $\forall$, $j$. For the second dose, since $\pi_{0,1} = 0.15$, this again means that we have little confidence that this dose is the MTD and, similarly, $\pi_{1,2}$ should still be the highest dose among $(\pi_{0,2}, \pi_{1,2}, \pi_{2,2})$. However, the probability of $\pi_{1,2}$ to be in $H_1$ should now be lower than that for $\pi_{1,1}$, since dose 2 has a higher toxic rate than dose 1. For example, if we assign the probabilities as $(\pi_{0,2} = 0.20, \pi_{1,2} = 0.60, \pi_{2,2} = 0.20)$, there should be a

**Table 1. Prior probabilities of each dose for three point hypotheses.**

| Priors | $D_1$ | $D_2$ | $\cdots$ | $D_j$ | $\cdots$ | $D_{d-1}$ | $D_J$ |
|---|---|---|---|---|---|---|---|
| $H_0$ | $\pi_{0,1}$ | $\pi_{0,2}$ | $\cdots$ | $\pi_{0,j}$ | $\cdots$ | $\pi_{0,J-1}$ | $\pi_{0,J}$ |
| $H_1$ | $\pi_{1,1}$ | $\pi_{1,2}$ | $\cdots$ | $\pi_{1,j}$ | $\cdots$ | $\pi_{1,J-1}$ | $\pi_{1,J}$ |
| $H_2$ | $\pi_{2,1}$ | $\pi_{2,2}$ | $\cdots$ | $\pi_{2,j}$ | $\cdots$ | $\pi_{2,J-1}$ | $\pi_{2,J}$ |
| Prob | 1 | 1 | 1 | 1 | 1 | 1 | 1 |

decreasing trend in the probability vector of $H_1$ and an increasing trend in the probability vector of $H_2$. Similarly, a decreasing trend will be observed for a vector probability of $H_2$.

Given the above premise, we propose the following algorithm that can automatically implement the assignment of horizontal probability vectors in Table 1. However, in reality, there will be infinite alternatives to elicit three probability vectors by satisfying the above increasing or decreasing monotone constraints. Our proposed alternative shows just one of the possible cases.

Step 1. Assign each dose a probability for $H_0$, that is, a prior probability vector of $(\pi_{0,1}, \cdots, \pi_{0,J})$, to best "guess" which of these $J$ doses to be the MTD.

This step is not so challenging if clinicians have strong confidence on which dose is closest to the MTD target. For example, in pediatric trials, we can often choose the MTD for adult patients or a starting MTD dose for pediatric patients. In this step, clinicians can also choose a set of skeletons for the CRM.

Step 2. If the dose $j$ is believed to be close to the MTD, then let $\frac{\pi_{1,j}}{\pi_{2,j}} = 1$, that is, odds$_j$ = odds $(\pi_{1,j}, \pi_{2,j}) = 1$ to assign probabilities to $\pi_{1,j}$ and $\pi_{2,j}$ given $\pi_{0,j}$ in Step 1. Also, let the lowest dose have odds$_1$ = odds$(\pi_{1,1}, \pi_{2,1}) = 10$ and the highest dose have odds$_J$ = odds$(\pi_{1,d}, \pi_{2,J}) = \frac{1}{10}$. We can have probabilities for $\pi_{1,1}, \pi_{2,1}$ and $\pi_{1,J}, \pi_{2,J}$. If the lowest or highest dose levels are believed to be the MTD, then the odds for it is set to be 1.

Step 3. Use extrapolation method (see details in Appendix) to elicit prior probabilities for the rest of two vectors $(\pi_{1,1}, \cdots, \pi_{1,J})$ and $(\pi_{2,1}, \cdots, \pi_{2,J})$ can be easily derived.

The above algorithm is easy to use since it only requires the investigator to provide probability guesses for $H_0$s for each investigated dose level. All the other remaining probabilities in Table 1 can be automatically computed, which substantially reduces the burden on investigators and improves the "guess" precision. See details of the algorithm and a numerical example to show the algorithm in the Appendix.

## 4 aBOIN design with adaptive boundaries

This section discusses the extension of the BOIN to aBOIN design with adaptive boundaries. For interval-based designs, the first step is to specify an indifference interval defined by two <u>fixed</u> boundaries to differentiate under-dose from over-toxic dose levels. Based on these boundaries, decision rules of dose assignment are developed. (See Introduction for the BOIN design.) The BOIN design is also categorized as a model-assisted design from the perspective of the modeling approach and how accumulative data are used [10]. The BOIN design derives two underlinefixed boundaries to make the dose escalation/de-escalation decision from its theories, denoted by $\lambda_1$ and $\lambda_2$, which are indirectly linked to the under- and over-dose hypotheses introduced earlier. If we denote the MTD toxicity rate as $\phi$ and use the authors' recommendation of $\phi_1 = 0.6\phi$ and $\phi_2 = 1.4\phi$, the two boundaries can be written as a function of $\phi_1$ and $\phi_2$.

The BOIN design also has useful theoretical properties, such as minimizing the decision-making error, long-term memory coherence, and convergence to the MTD dose. In this section, we first demonstrate that the proposed aBOIN design also inherits theoretical properties from the BOIN design and then conduct simulation studies to see whether this extension could improve the original BOIN design.

## 4.1 Adaptive BOIN design with shrinking boundaries

Extensive simulation studies have shown that the BOIN design is simple but has excellent operating characteristics comparable with those of the more complicated model-based CRM designs [12].

Adaptive shrinking boundaries can possibly be used to further control the misallocation of patients to over-toxic doses. In the BOIN design framework, we reconstruct the $\phi_1$ and $\phi_2$ to be $\phi_1 = \phi - \frac{\Delta_1}{(\sqrt{n_j})^{g_1}}$ and $\phi_2 = \phi + \frac{\Delta_2}{(\sqrt{n_j})^{g_2}}$. Here, $n_j$ is the cumulative number of patients treated at a dose level of $j$ during the trial and $0 < g_1, g_2 < 1$ are *discounting parameters* to control the shrinking speed of the two boundaries. Parameters $\Delta_1, \Delta_2$ can be interpreted as pre-specified effect sizes to construct the decision intervals in the BOIN design as given above. Obviously, by doing so, the two fixed boundaries of the original BOIN design now depend on the dynamic number $n_j$s, which is number of patients treated at the dose level $j$. This way of construction would clearly make $\phi_1$ and $\phi_2$ converge to the MTD target $\phi$ as $\phi_1$ increases to $\phi$; in other words, the interval $(\phi_1, \phi_2)$ is bound to converge to the MTD as sample sizes increase. This construction is also very flexible for designing trials. For example, if safety of the design is a very big concern, we can make the upper boundary $\phi_2$ to shrink faster than the lower boundary $\phi_1$ by using discounting factors $g_1 < g_2$ to penalize assignment of patients to dose levels beyond the MTD.

Based on the above redefinition of $\phi_1$ and $\phi_2$, we have the updated three-point hypotheses of the BOIN design as

$$H_{0j} : p_j = \phi,$$

$$H_{1j} : p_j = \phi_1 = \phi - \frac{\Delta_1}{(\sqrt{n_j})^{g_1}},$$

and

$$H_{2j} : p_j = \phi_2 = \phi + \frac{\Delta_2}{(\sqrt{n_j})^{g_2}},$$

The above definition of $\phi_1$ and $\phi_2$ is reminiscent of boundaries of an optimal symmetric group sequential design by Eales & Jennison (1992) [13]. However, they are primarily used to ensure type I and type II error rates for confirmatory trials.

In a similar vein of deviations for the BOIN design, the optimal $\lambda_{1j}$ and $\lambda_{2j}$ minimize the decision error rate can be derived as $\lambda_1(\Delta_1, n_j)$ and $\lambda_2(\Delta_2, n_j)$ with $\phi_1 = \phi - \frac{\Delta_1}{(\sqrt{n_j})^{g_1}}, \phi_2 = \phi + \frac{\Delta_2}{(\sqrt{n_j})^{g_2}}$ plugging into (1) and (2), respectively:

$$\lambda_1(\Delta_1, n_j) = \frac{\log\left(1 + \frac{\Delta_1}{(1-\phi)\sqrt{n_j^{g_1}}}\right)}{\log\left\{\frac{\left(1 + \frac{\Delta_1}{(1-\phi)\sqrt{n_j^{g_1}}}\right)}{\left(1 - \frac{\Delta_1}{\phi\sqrt{n_j^{g_1}}}\right)}\right\}} \tag{5}$$

$$\lambda_2(\Delta_2, n_j) = \frac{\log^{-1}\left(1 - \frac{\Delta_2}{\sqrt{n_j^{g_2}}(1-\phi)}\right)}{\log\left\{\frac{\left(1 + \frac{\Delta_2}{\phi\sqrt{n_j^{g_2}}}\right)}{\left(1 - \frac{\Delta_2}{(1-\phi)\sqrt{n_j^{g_2}}}\right)}\right\}} . \tag{6}$$

Different from the original BOIN design, $\lambda_1(\Delta_1, n_j)$ and $\lambda_2(\Delta_2, n_j)$ is presently depend on accumulative sample size $n_j$ along the trial process instead of constants. We can show that the aBOIN design still enjoys the following theoretical properties

**Theorem 1**. *The proposed aBOIN design has (1) long-term memory coherence, (2) converges to the MTD, and (3)* $\lambda_1 < \phi, \lambda_2 > \phi$.

See Appendix for proofs.

## 4.2 Practical implementation of the adaptive BOIN design

To use the proposed aBOIN design in practice, we need to specify the values of $\Delta_1$ and $\Delta_2$, which determine the $\phi_1$, $\phi_2$ and subsequently the values of $\lambda_1$, $\lambda_2$. Since the original BOIN design recommends $\phi_1 = 0.6\phi$ and $\phi_2 = 1.4\phi$, we recommend that $\Delta_1 = \Delta_2 = 0.4\phi$, which is exactly the same as the original BOIN design when $n_j = 1$.

In practice, we also introduce a lead-in process in which we follow the procedure given in the original BOIN design for a pre-specified number of patients, for example, $N_1$, and the trial then switches to the aBOIN design with adaptive shrinking boundaries.

Our exploratory simulations (not shown here) with a maximum sample size of 30 show negligible differences in the performance of the trial when $N_1 = 6$ or $N_1 = 9$ is used. Hereafter, we will use $N_1 = 6$ in simulation studies for the lead-in period.

Note that by adopting the accelerating parameter $g_1$ and $g_2$, hypotheses of $H_{1j}$ and $H_{2j}$ are no longer symmetric. However, including accelerating parameters $g_1$ and $g_2$ does not influence the asymptotic properties of the aBOIN design. Furthermore, different $g_1$ and $g_2$ may satisfy practical needs; for example, if we want a tighter control of the over toxicities, we can let $g_2 > g_1$, which means that the upper boundary would shrink quicker than the below boundary.

Additionally, the aBOIN design that incorporates external information can be derived straightforwardly to have the following form:

$$\lambda_{1j}(n_j) = \frac{\log\left(1 + \frac{\Delta_1}{(1-\phi)(\sqrt{n_j})^{g_1}}\right) + n_j^{-1}\log\left(\frac{\pi_{1j}}{\pi_{0j}}\right)}{\log\left\{\frac{\left(1 + \frac{\Delta_1}{(1-\phi)(\sqrt{n_j})^{g_1}}\right)}{\left(1 - \frac{\Delta_1}{\phi(\sqrt{n_j})^{g_1}}\right)}\right\}} \tag{7}$$

$$\lambda_{2j}(n_j) = \frac{\log^{-1}\left(1 - \frac{\Delta_2}{(\sqrt{n_j})^{g_2}(1-\phi)}\right) + n_j^{-1}\log\left(\frac{\pi_{0j}}{\pi_{2j}}\right)}{\log\left\{\frac{\left(1 + \frac{\Delta_2}{\phi(\sqrt{n_j})^{g_2}}\right)}{\left(1 - \frac{\Delta_2}{(1-\phi)(\sqrt{n_j})^{g_2}}\right)}\right\}} \tag{8}$$

## 5 Simulation studies

In this section, we explore the operating characteristics of the proposed aBOIN design with and without incorporating prior information by comparing it to the original BOIN design. The aims of the simulation study are twofold: (i) to explore the behavior of the aBOIN design that incorporates prior information compared with that of the original BOIN design and the

**Table 2. Ten true toxicity scenarios with the target DLT rate of 20% and 30%.**

| Scenario | Dose Level | | | | | | | | | |
|---|---|---|---|---|---|---|---|---|---|---|
| | DLT 20% | | | | | DLT 30% | | | | |
| | 1 | 2 | 3 | 4 | 5 | 1 | 2 | 3 | 4 | 5 |
| 1 | **0.20** | 0.22 | 0.23 | 0.25 | 0.27 | **0.30** | 0.33 | 0.34 | 0.35 | 0.36 |
| 2 | 0.18 | **0.20** | 0.22 | 0.23 | 0.25 | 0.27 | **0.30** | 0.33 | 0.34 | 0.35 |
| 3 | 0.17 | 0.18 | **0.20** | 0.22 | 0.23 | 0.26 | 0.27 | **0.30** | 0.33 | 0.34 |
| 4 | 0.1 | 0.15 | 0.18 | **0.20** | 0.22 | 0.15 | 0.2 | 0.27 | **0.30** | 0.33 |
| 5 | 0.08 | 0.1 | 0.15 | 0.18 | **0.20** | 0.1 | 0.15 | 0.2 | 0.27 | **0.30** |
| 6 | **0.20** | 0.3 | 0.35 | 0.4 | 0.45 | **0.30** | 0.4 | 0.45 | 0.5 | 0.55 |
| 7 | 0.1 | **0.20** | 0.3 | 0.35 | 0.4 | 0.2 | **0.30** | 0.4 | 0.45 | 0.5 |
| 8 | 0.05 | 0.1 | **0.20** | 0.3 | 0.35 | 0.1 | 0.2 | **0.30** | 0.4 | 0.45 |
| 9 | 0.01 | 0.05 | 0.1 | **0.20** | 0.3 | 0.05 | 0.1 | 0.2 | **0.30** | 0.4 |
| 10 | 0.01 | 0.05 | 0.08 | 0.1 | **0.20** | 0.05 | 0.1 | 0.15 | 0.2 | **0.30** |

aBOIN design that does not incorporate prior information, and (ii) explore the operating characteristics of the original BOIN and aBOIN designs.

**Simulation setting**

We consider trials with five dose levels and a maximum sample size of 30 patients, with a cohort size of three patients. Twenty different scenarios (one half with dose-limiting toxicity (DLT) rates of 20%, and the other half with DLT rates of 30%) with various locations and DLT rates are shown in Table 2. We will use them to examine properties of the proposed designs.

For each scenario, we simulated 10,000 trials. We implemented the BOIN design using the R package BOIN with its default design parameters. For the aBOIN design, we specified the accelerating factors as $g_1 = 0.4$, $g_2 = 0.9$, which were derived by trial and error, and we only activated the adaptive shrinking mechanism in at least six patients who had been treated (referred to as the lead-in period). As introduced in [10], four metrics to measure the performance of a design have been considered: (1) the percentage of correct selection (PCS) of the true MTD in 10,000 simulated trials; (2) the average number of patients allocated to the MTD across 10,000 simulated trials; (3) the risk of overdosing, defined as the percentage of simulated trials in which a large percentage (e.g., more than 60% or 80%) of patients are treated at doses above the MTD (i.e., how likely it is that the design treats more than 60% or 80% of patients at doses above the MTD); and (iv) the risk of under-dosing, which is defined as the percentage of simulated trials in which more than 80% of patients are treated at doses below the MTD (potential sub-therapeutic doses). For comparing the aBOIN design with or without prior information, we focus on the PCS of the true MTD by comparing the proposed design to the original BOIN design.

## 5.1 Simulation 1: Adaptive BOIN design with incorporating prior information

To incorporate prior information, we first specify a probability vector for $H_0$ row in Table 1. In our simulations, we assign one set of probability vectors for all 20 scenarios in Table 2, because this enables us to check whether the performance of the aBOIN design with prior information is robust or not through various locations of MTDs in Table 2. To be specific, we assign $H_0$ with probabilities $(\pi_{0,1}, \cdots, \pi_{0,5}) = (0.2, 0.45, 0.7, 0.45, 0.2)$ for all scenarios, and the other two probability vectors are derived by using the procedure introduced above to be $(\pi_{1,1}, \cdots, \pi_{1,5}) = (0.72, 0.44, 0.15, 0.12, 0.08)$ and $(\pi_{2,1}, \cdots, \pi_{2,5}) = (0.08, 0.11, 0.15, 0.43, 0.72)$. Considering

**Table 3. Percentage of correctly selection percentage of MTD for ten scenarios in Table 1.**

| Scenario | 1 | 2 | 3 | 4 | 5 | 6 | 7 | 8 | 9 | 10 | Average |
|---|---|---|---|---|---|---|---|---|---|---|---|
| DLT rate 20% | | | | | | | | | | | |
| PCS(%) | | | | | | | | | | | |
| BOIN | 37.15 | 22.81 | 16.1 | 16.38 | 27.3 | 55.11 | 48.96 | 45.41 | 46.27 | 56.69 | 37.22 |
| aBOIN[1] | 32.22 | 20.67 | 16.75 | 18.17 | 34.33 | 51.08 | 46.32 | 43.82 | 46.63 | 60.05 | 37.0 |
| aBOIN[2] | 26.03 | 23.39 | 22.61 | 14.99 | 33.12 | 51.07 | 45.2 | 47.65 | 39.1 | 57.28 | 36.04 |
| # of Patients at MTD | | | | | | | | | | | |
| BOIN | 16.81 | 8.1 | 4.78 | 4.02 | 4.98 | 20.4 | 13.62 | 11.22 | 10.11 | 10.93 | 10.49 |
| aBOIN | 14.48 | 8.13 | 5.49 | 4.89 | 6.67 | 18.24 | 12.82 | 10.81 | 10.15 | 11.92 | 10.36 |
| Risk of overdosing 60% | | | | | | | | | | | |
| BOIN | 31.44 | 16.53 | 6.59 | 3.71 | 0 | 16.84 | 10.69 | 6.77 | 3.67 | 0 | 9.62 |
| aBOIN | 40.5 | 23.63 | 9.77 | 2.59 | 0 | 22.85 | 14.81 | 7.46 | 2.14 | 0 | 12.38 |
| Risk of underdosing 60% | | | | | | | | | | | |
| BOIN | 0 | 32.67 | 55.45 | 70.38 | 80.93 | 0 | 23.82 | 35.38 | 41.35 | 59.09 | 39.91 |
| aBOIN | 0 | 24.86 | 43.54 | 60.42 | 74.36 | 0 | 19.85 | 29.79 | 33.25 | 53.75 | 33.98 |
| DLT rate 30% | | | | | | | | | | | |
| PCS(%) | | | | | | | | | | | |
| BOIN | 41.08 | 24.5 | 18.8 | 21.57 | 38.92 | 55.17 | 45.65 | 43.29 | 43.92 | 59.06 | 39.2 |
| aBOIN[1] | 39.08 | 26.3 | 19.7 | 21.16 | 36.14 | 54.1 | 46.52 | 41.51 | 42.37 | 56.85 | 38.37 |
| aBOIN[2] | 31.01 | 27.29 | 25.67 | 19.97 | 39.64 | 53.23 | 46.25 | 45.31 | 38.39 | 58.8 | 38.56 |
| # of Patients at MTD | | | | | | | | | | | |
| BOIN | 17.66 | 9 | 5.49 | 5.4 | 7.25 | 20.47 | 13.4 | 11.16 | 10.01 | 11.66 | 11.15 |
| aBOIN | 17.58 | 9.19 | 5.59 | 4.97 | 5.97 | 20.32 | 12.58 | 10.17 | 9.21 | 9.68 | 10.53 |
| Risk of overdosing 60% | | | | | | | | | | | |
| BOIN | 35.29 | 18.63 | 6.25 | 1.92 | 0 | 23.08 | 15.19 | 8.55 | 2.17 | 0 | 11.11 |
| aBOIN | 29.36 | 12.01 | 4.1 | 0.78 | 0 | 16.73 | 8.24 | 4.71 | 0.78 | 0 | 7.67 |
| Risk of underdosing 60% | | | | | | | | | | | |
| BOIN | 0 | 33.07 | 54.08 | 62.45 | 71.54 | 0 | 23.77 | 33.04 | 37.28 | 52.5 | 36.77 |
| aBOIN | 0 | 29.07 | 56.42 | 70.05 | 82.23 | 0 | 21 | 38.26 | 45.8 | 68.17 | 41.1 |

aBOIN[1]: adaptive BOIN design without incorporating prior information.

aBOIN[2]: adaptive BOIN design with incorporating prior information.

these specific settings, this means that we have 70% confidence that dose level 3 could be the MTD and 45% confidence that dose level 2 or 4 could be the MTD.

Simulation results of the PCS (%) for the BOIN and the aBOIN design with or without incorporating prior information are shown in Table 3. For example, with a DLT rate of 20% for the PCS (%) metric, there are 10 scenarios. The first row with "BOIN" refers to the PCS (%) of the original BOIN design. For instance, for scenarios 1, 3, and 8, the corresponding PCSs (%) are 37.15%, 16.1%, and 45.41% with MTD locations at dose levels 1, 3, and 3, respectively. The second row with aBOIN[1] refers to PCS (%) of the aBOIN design without incorporating prior information. Similarly, for scenarios 1, 3, and 8, the corresponding PCSs (%) are 32.22%, 16.75%, and 43.82% with MTD locations at dose levels 1, 3, and 8, respectively. The third row with aBOIN[2] refers to the PCS (%) of the aBOIN design incorporating prior information. The corresponding PCS (%) for scenarios 1, 3, and 8 are 26.03%, 22.61%, and 47.65%, respectively. Because the prior has placed high confidence on dose level 3 being the MTD, and scenarios 3 and 8 are scenarios with MTD locations at dose level 3, for scenarios 3 and 8, aBOIN[2] has

highest PCSs (%) among the three designs. For the remaining scenarios, the PCS (%) of aBOIN[2] is comparable to or lower than that of the BOIN and aBOIN[1] designs. For example, for scenario 3 of DLT with a 20% toxicity rate, the PCS (%) of the aBOIN[2] is 22.61%, whereas those of the BOIN and aBOIN[1] are 16.1% and 16.75%, respectively, because for scenario 3 with the MTD located at dose level 3, the prior guess also has the strongest confidence at dose level 3. However, if we check scenario 1 with the MTD located at dose level 1, we put only 20% confidence into dose level 1 and find that the aBOIN[2] design has the worst performance in terms of the PCS% (26.03%), whereas BOIN and aBOIN[1] have a higher PCS% (37.15% and 32.22%, respectively). At a DLT rate of 30%, similar patterns can be observed.

From these results, we infer that when the prior guess for the MTD location is close to the truth, the aBOIN version incorporating prior information performs the best in terms of the PCS metric; in other scenarios, its results vary widely and can sometimes even be very inaccurate. Given these observations, we recommend that in actual practice, the aBOIN incorporating prior information should be used only when the investigator has strong confidence or there is prior or historical information on which dose is or approximate to the MTD.

## 5.2 Simulation 2: Adaptive BOIN design without incorporating prior information

In this subsection, we investigate the aBOIN[1] design, that is, the aBOIN design without incorporating prior information. However, in this subsection, we still call this version aBOIN for brevity. We closely examine not only the PCS% but also the other three metrics, percentages of patients allocated to a true MTD during the trial (MTD%), and the mean number of observed DLTs throughout the trial (# of DLTs). Results are shown in Table 3 and Figs 1 and 2.

**Results.** For a DLT rate of 20%, Table 3 and Fig 1 show that the PCS (%) of the aBOIN design for scenarios such as 1, 2, 6, 7, 8, and 10 is comparable to or lower than that for the BOIN design. For the remaining scenarios, performance of the aBOIN design by PCS (%) as a metric is comparable with that of the original BOIN design. Findings are similar for the criterion "# of Patients at MTD." However, for the criterion "Risk of Overdosing 60 (%)," in almost across all scenarios, the risks associated with the aBOIN design are higher than for the BOIN design. Nevertheless, for the criterion "Risk of Underdosing 60 (%)," the BOIN design performs poorer than the aBOIN design. For a DLT rate of 30%, Table 3 and Fig 1 show that the BOIN and aBOIN designs have comparable performances for the criteria "PCS (%)" and "# of Patients at MTD." However, at a DLT rate of 20%, for the criterion "Risk of Overdosing 60 (%)," overall the aBOIN design is associated with lower risks than for the BOIN design across all scenarios but higher risks than that for the BOIN design for the criterion "Risk of Underdosing 60(%)."

We also examined the convergence rate with the PCS (%) metric for asymptomatic properties for both designs. We present partial results for the first four scenarios for DLT rates of 20% and 30%. Fig 3 shows that for DLT rates of 20%, in all explored scenarios the curve of the aBOIN design is eventually above that of the original BOIN design. This indicates that as the sample size increases, the aBOIN design performs better than the BOIN design in terms of the PCS metric. Although we know that phase I trials usually have small sizes (approximately 30 patients), asymptotic findings show that the idea of shrinking boundaries comes into effect only when the sample size is larger, and therefore this idea has little practical use although it is asymptotically or theoretically meaningful. In summary, for the finite sample, the aBOIN and BOIN designs have comparable performances with respect to the four criteria. For large samples, the aBOIN design performs better than the original BOIN design, but it has little practical

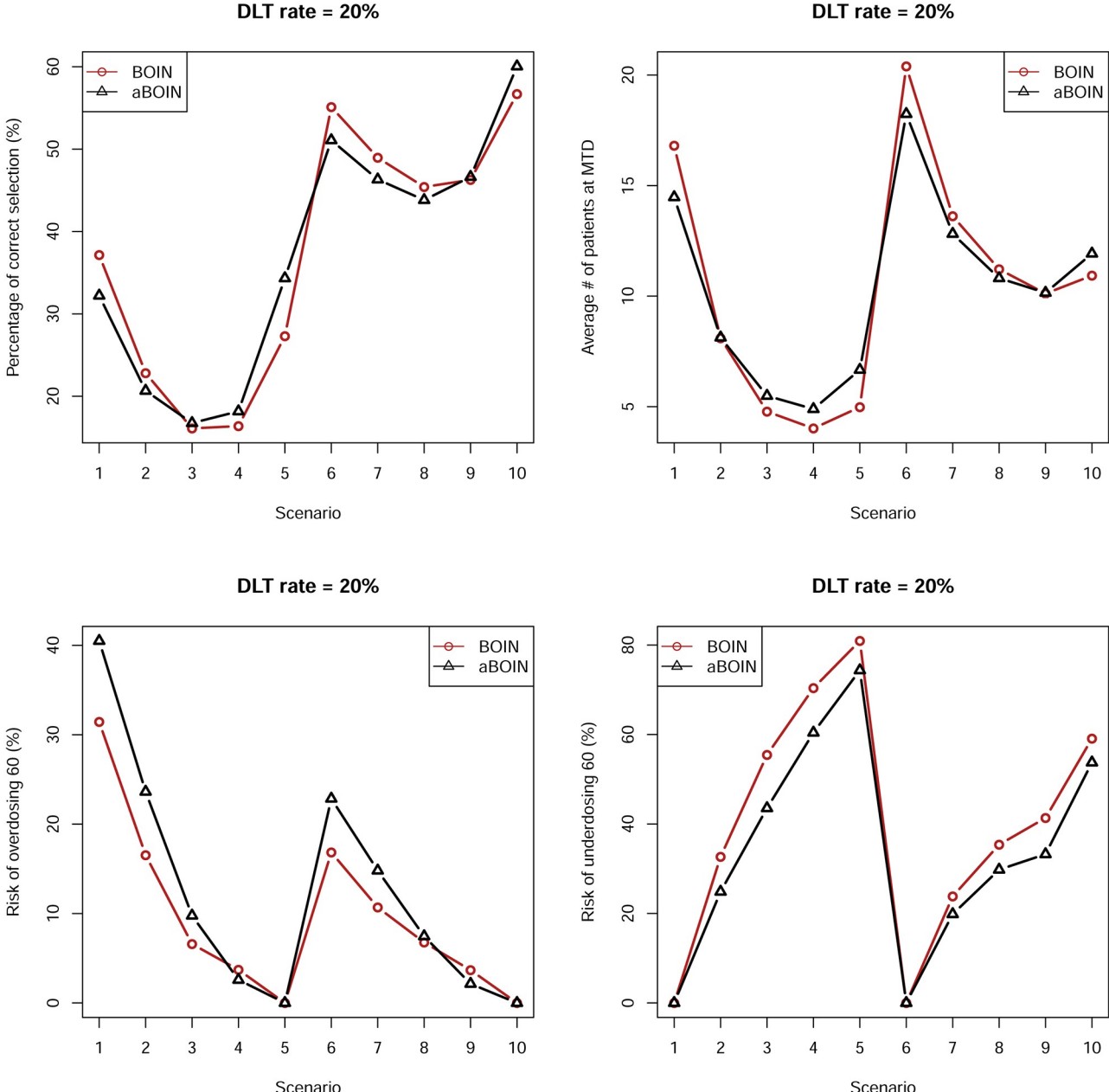

**Fig 1. Operating characteristics of ten scenarios on the left panel (DLT 20%) of Table 2 by two competing methods BOIN and aBOIN.**

use due to limited sample size in practice, and the corresponding simulated results can be seen in Fig 4.

Note that if investigators have vague or less confidence about prior experience or information, we still suggest that they use the BOIN design without prior information.

## 6 Discussion

We have developed two extensions of the BOIN design. The first one develops an accessible approach to allow the incorporation of prior or historical information in the phase I trial. The

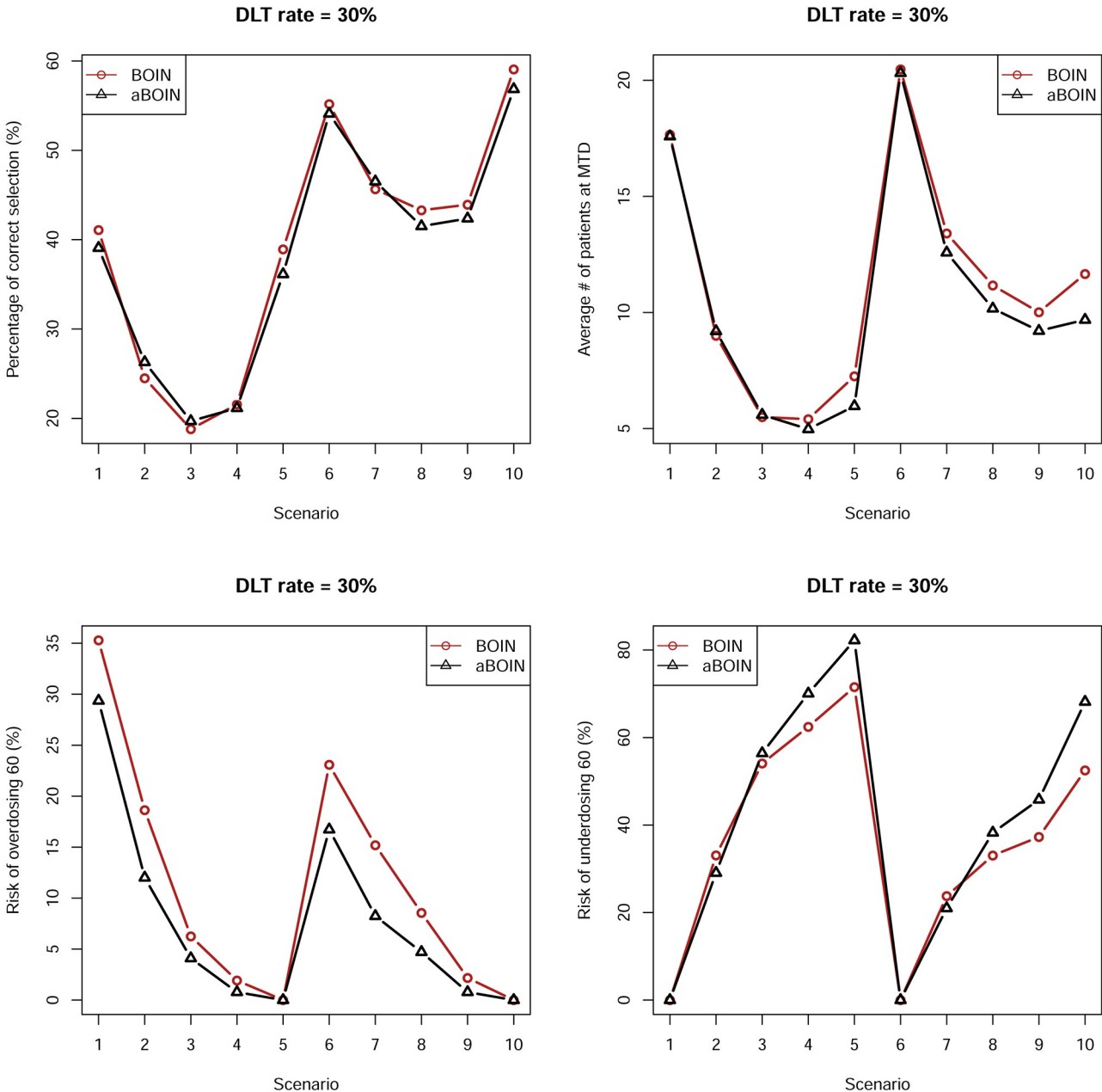

**Fig 2. Operating characteristics of ten scenarios on the left panel (DLT 30%) of Table 2 by two competing methods BOIN and aBOIN.**

second extension proposes adaptive shrinking boundaries (aBOIN design), whereas the original BOIN design has fixed boundaries. The aBOIN design uses accelerating factors to control the shrinking speed rates of lower and upper boundaries. Theoretical properties were derived for the aBOIN design.

Performances of the proposed methods were discussed by simulations. When setting up the location for the MTD a priori that was close to the MTD, the aBOIN design incorporating prior information showed better performance than the original BOIN design. However, if the prior deviated from the truth, performance of the aBOIN design was inferior to that of the BOIN design. This is understandable, since there were very few sample sizes and therefore it

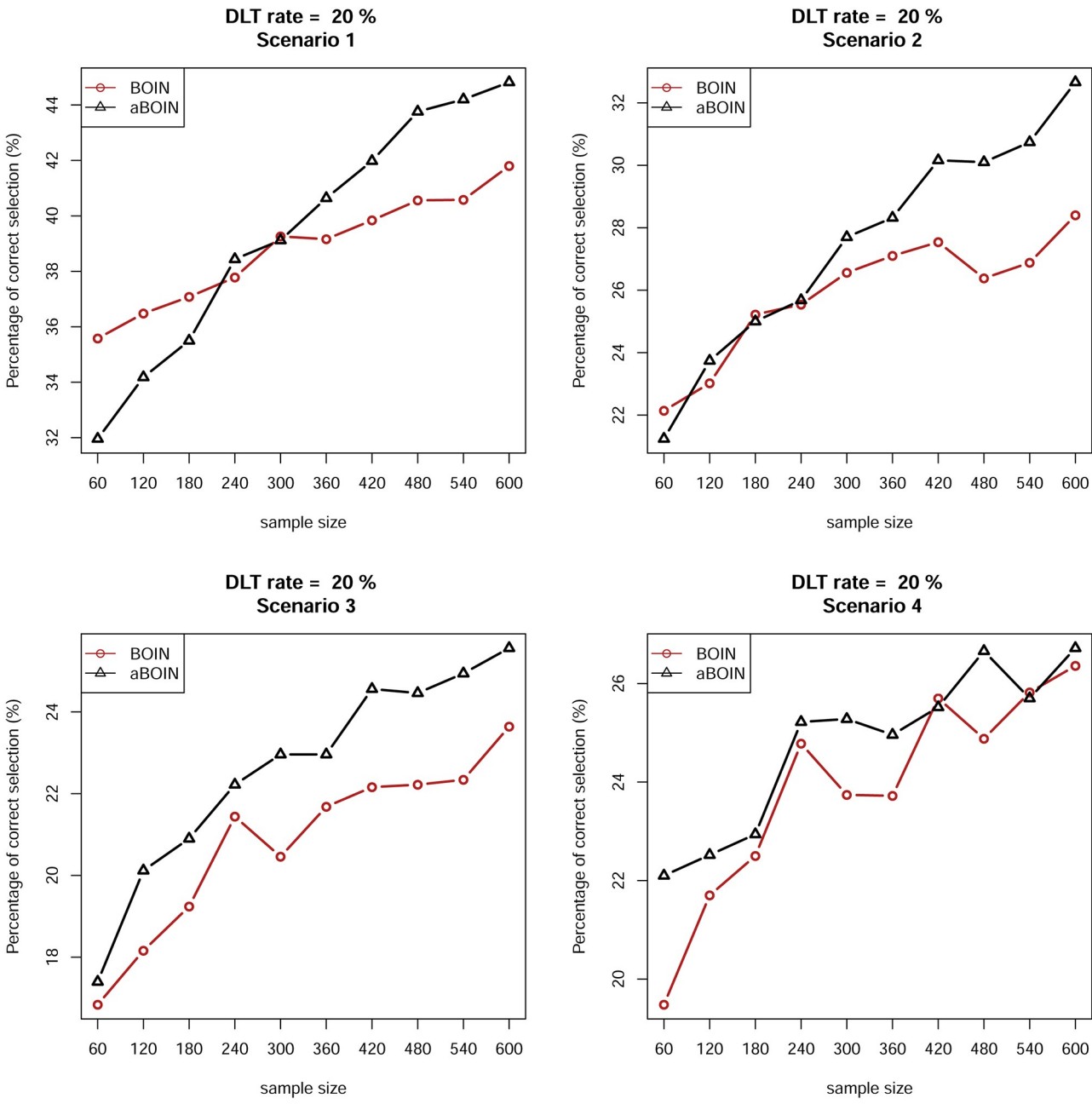

**Fig 3. Percentage of correctly selection percentage of MTD of the first four scenarios on the left panel (DLT rate 20%) in Table 2 by using the two competing methods BOIN and aBOIN with a large sample.**

was hard to dominate the estimation procedure for deciding the dose. Therefore, we caution practitioners to use prior information in real trials unless there is strong confidence. The second extension of the proposed aBOIN design was examined numerically by using a finite sample and a large sample. For finite sample sizes, performances were similar when comparing the aBOIN without incorporating prior information to the BOIN design. Although the proposed aBOIN design outperforms in asymptotic properties, it has limited use in actual phase I trials due to the small sample size. In summary, the original BOIN design can be improved only if very informative historical information is available.

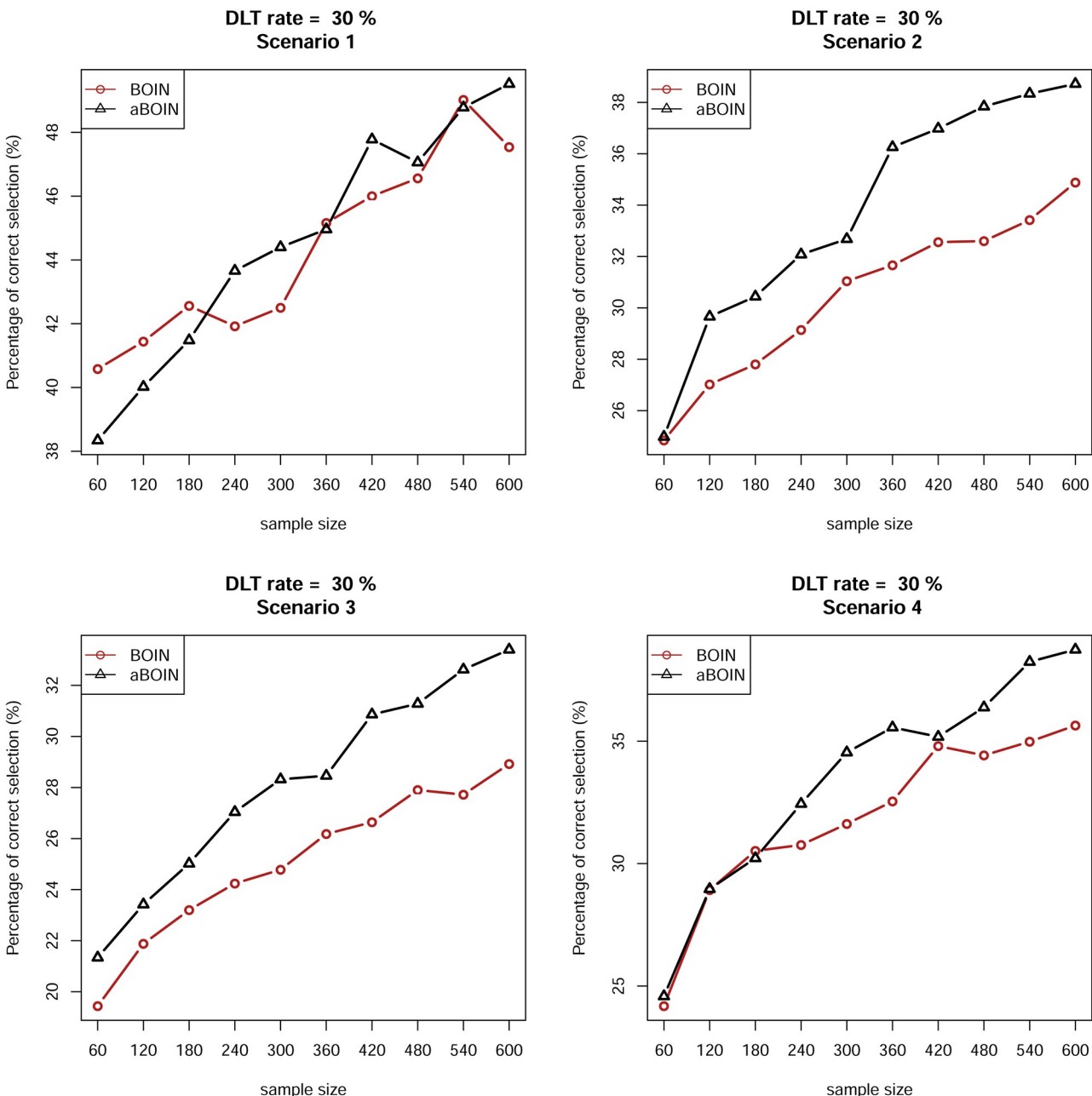

**Fig 4. Correctly selected percentage of the maximum tolerated dose for the first four scenarios in the right panel at a DLT rate of 30% in Table 2 by using original BOIN and aBOIN designs in a large sample.**

## Appendix 1: Proof of theoretical properties

*Proof. Coherence.* Since $\lambda_{1j} < \phi$ and $\lambda_{2j} > \phi$, we can easily obtain the coherence:

$$\text{pr}(\text{dose escalation}|\hat{p}_j > \phi) = \text{pr}(\hat{p}_j < \lambda_{1j}|\hat{p}_j > \phi) = 0,$$

$$\text{pr}(\text{dose deescalation}|\hat{p}_j < \phi) = \text{pr}(\hat{p}_j > \lambda_{1j}|\hat{p}_j < \phi) = 0.$$

Thus, the aBOIN is long-term memory coherent.

*Proof* By the definition of $\phi_1$ and $\phi_2$, we can get $\lambda_1 \to \phi$, $\lambda_2 \to \phi$, as $n_j$ tends to $\infty$. By the L'hopital's rule, we get

$$
\begin{aligned}
\lim_{\phi_1 \to \phi} \lambda_1 &= \lim_{\phi_1 \to \phi} \log\left(\frac{1-\phi_1}{1-\phi}\right) / \log\left\{\frac{\phi(1-\phi_1)}{\phi_1(1-\phi)}\right\} \\
&= \lim_{\phi_1 \to \phi} \frac{\frac{-1}{1-\phi_1}}{\frac{-\phi}{\phi(1-\phi_1)} - \frac{1-\phi}{\phi_1(1-\phi)}} = \lim_{\phi_1 \to \phi} \frac{\frac{-1}{1-\phi_1}}{\frac{-1}{1-\phi_1} - \frac{1}{\phi_1}} \\
&= \lim_{\phi_1 \to \phi} \frac{-\phi_1}{-\phi_1 - 1 + \phi_1} = \lim_{\phi_1 \to \phi} \phi_1 \to \phi
\end{aligned}
\tag{9}
$$

The proof of $\lambda_2 \to \phi$ as $n_j \to \infty$ is similar as above.

That is, both $\lambda_1$ and $\lambda_2$ shrink toward the MTD target $\phi$.

*Proof.* $\lambda_1 < \phi$ and $\lambda_2 > \phi$). Prove $\lambda_1 < \phi$.

Since we have proved that $\lambda_1$ converges to $\phi(>0)$, if we prove $\lambda_1$ is an increasing function of $\phi_1$, then we can prove $\lambda_1 < \phi$. Let $\lambda_1 = f(\phi_1) = \frac{\log\left(\frac{1-\phi_1}{1-\phi}\right)}{\log\left\{\frac{\phi(1-\phi_1)}{\phi_1(1-\phi)}\right\}}$ and $\frac{df(\phi_1)}{d\phi_1} =$

$\frac{\frac{-1}{1-\phi_1}[\log\phi + \log(1-\phi_1) - \log\phi_1 - \log(1-\phi) - [\log(1-\phi_1) - \log(1-\phi)][\frac{-1}{1-\phi_1} - \frac{1}{\phi_1}]}{[\log\phi + \log(1-\phi_1) - \log\phi_1 - \log(1-\phi)]^2}$ The numerator $=$

$\frac{\log\phi_1 - \log\phi + 1 - \phi_1}{(1-\phi_1)\phi_1} > \frac{\frac{\phi_1-1}{\phi_1} - \frac{\phi-1}{\phi} + 1 - \phi_1}{(1-\phi_1)\phi_1} = \frac{1-\phi+\phi_1\phi}{(1-\phi_1)\phi_1^2\phi} > 0$ (using inequality $\log(x) > \frac{x-1}{x}$ for all $x > -1$)

(since $0 < \phi, \phi_1 < 1$)

Thus, $\frac{df(\phi_1)}{d\phi_1} > 0$, that is, $\lambda_1 = f(\phi_1)$ is an increasing function of $\phi_1$ with limit at $\phi$.

Hence, we have $\lambda_1 < \phi$.

Similarly, we can prove $\lambda_2 > \phi$.

*Proof. Convergency.* Denote the event $A = \hat{p}_j \in (\lambda_1, \lambda_2)$. We only need to show that when $n_j$ is large enough, $\mathrm{pr}(A) = 1$. Since $\lim_{nj \to \infty} \lambda_1/\phi_1 = 1$ and $\lim_{nj \to \infty} \lambda_2/\phi_2 = 1$, for $g_1, g_2 < 1$ by the CLT, we can get

$$
\begin{aligned}
\mathrm{pr}(\lim_{n_j \to \infty} A) &= \mathrm{pr}(\lim_{n_j \to \infty} \hat{p}_j \in (\lambda_1, \lambda_2)) = \mathrm{pr}(\lim_{n_j \to \infty} \hat{p}_j \in (\phi_1, \phi_2)) \\
&= \mathrm{pr}(\lim_{n_j \to \infty} \phi - \frac{\Delta_1}{(\sqrt{n_j})^{g_1}} < \hat{p}_j < \phi + \frac{\Delta_2}{(\sqrt{n_j})^{g_2}}) \\
&= \mathrm{pr}(\lim_{n_j \to \infty} -\frac{\Delta_1\sqrt{n_j}}{(\sqrt{n_j})^{g_1}\phi(1-\phi)} < \frac{\sqrt{n_j}(\hat{p}_j - \phi)}{\phi(1-\phi)} < \frac{\Delta_2\sqrt{n_j}}{(\sqrt{n_j})^{g_2}\phi(1-\phi)}) \\
&= \Phi(\infty) - \Phi(-\infty) = 1.
\end{aligned}
$$

Then, the proof provided by Oron, Azriel, and Hoff (2011) can be directly used to obtain the result.

## Appendix 2: Algorithm of generating priors in Table 1

Assuming $J$ dose levels, we firstly elicit a prior vector probability for hypothesis $H_0$, that is, guessing which dose would possibly be the MTD, denoted as $\pi_{0,1}, \cdots \pi_{0,j}, \cdots, \pi_{0,J}$. We can also assume odds of $H_1$ to $H_2$ at dose level 1, since at the lowest dose, it would have high confidence that this first dose would be under-dosing than over-dosing, thus, we let $\mathrm{odds}_1 = \frac{\pi_{1,1}}{\pi_{2,1}}$ can be any large number, for instance, $\mathrm{odds}_1 = \frac{\pi_{1,1}}{\pi_{2,1}} = 10$ in our algorithm, and, vice versa, the odds of $H_1$ to $H_2$ at dose level $J$, would be a small number, for instance, we let $\mathrm{odds}_J \frac{\pi_{1,J}}{\pi_{2,J}} = \frac{1}{10}$. If dose level $j$ is assigned highest probability, that is, dose $j$ is believed to be closet to the MTD prior to

the study and we assume there has equal chance to be under- or over-dose at this dose level, that is, $\text{odds}_J \frac{\pi_{1,j}}{\pi_{2,j}} = 1$.

Based on the above $\text{odds}_1$ and $\text{odds}_J$, we use the definition of the odds to evaluate the prior probabilities for $H_1$ at dose levels 1 and J as:

$$\pi_{1,j^*} = \frac{(1 - \pi_{0,j^*}) \times odds_{j^*}}{1 + odds_{j^*}}, j^* = 1 \ or \ J \tag{10}$$

then, the prior probabilities for $H_2$ at dose levels 1 and J are:

$$\pi_{J,j^*} = 1 - \pi_{1,j^*} - \pi_{0,j^*}, j^* = 1 \ or \ J \tag{11}$$

Now, we have the prior probabilities of the first row for $H_0$ and first and last (J-th) columns in Table 1. Based on these information, we then use an interpolation technique to assign probabilities for the rest cells in Table 1.

To be specific, for computing probabilities of $H_1$ for dose levels from 2 to $j - 1$, the following linear interpolation formula is used:

$$\pi_{1,j'} = \frac{(\pi_{1,1} + \pi_{1,j})}{j'}, 2 \leq j' \leq j - 1 \tag{12}$$

For computing probabilities of $H_1$ for dose levels from $j + 1$ to $J$, the following linear interpolation formula is used:

$$\pi_{1,j'} = \frac{(\pi_{1,j} + \pi_{1,J})}{j'}, j + 1 \leq j' \leq J \tag{13}$$

Thus, based on the above steps, we assign probabilities for the first ($H_0$) and second rows ($H_1$) in Table 1. Probabilities for third row ($H_2$) are:

$$\pi_{2,j'} = 1 - \pi_{0,j'} - \pi_{1,j'}, 1 \leq j' \leq J \tag{14}$$

We provide a numerical example for showing the above procedure. Assuming there are 5 dose levels and, without losing generality, assuming the 3rd dose level is closest to the MTD prior to the study. For example, the prior probability vector is set to be $(\pi_{0,1}, \cdots, \pi_{0,5}) = (0.2, 0.45, 0.7, 0.45, 0.2)$, that is, this is the 1st row in Table 1. To be specific, we think that the dose level 3 may be close to the MTD with 70% confidence, and then dose level 2 and 4 with 45% confidence to be the MTD while the first dose level and last dose level have the minimal confidence to be the MTD with 20% for each. Since the odds (defined by the algorithm) for the dose level 3 is $\text{odds}_{3=} 1$, so we have $\pi_{1,3} = \pi_{2,3} = 0.15$ since $\pi_{0,3} = 0.7$ now. By the algorithm, we also know that odds for the first and last dose levels are as $\text{odds}_1 = 10$ and $\text{odds}_5 = \frac{1}{10}$.

By using the above formula (10) and (11), we can have

$$\pi_{1,1} = \frac{(1 - \pi_{0,1}) \times odds_1}{1 + odds_1} = \frac{(1 - 0.2) \times 10}{1 + 10} = 0.72,$$

$$\pi_{1,5} = \frac{(1 - \pi_{0,5}) \times odds_5}{1 + odds_5} = \frac{(1 - 0.2) \times 1/10}{1 + 1/10} = 0.08,$$

Then, we can have $\pi_{2,1} = 1 - \pi_{1,1} - \pi_{0,1} = 1 - 0.72 - 0.2 = 0.08$ and $\pi_{2,5} = 1 - \pi_{1,5} - \pi_{0,5} = 1 - 0.08 - 0.2 = 0.72$.

Thus, by using the formula (13) and (14), we have

$$\pi_{1,2} = \frac{(\pi_{1,1} + \pi_{1,3})}{2} = \frac{(0.72 + 0.15)}{2} = 0.44$$

$$\pi_{1,4} = \frac{(\pi_{1,3} + \pi_{1,5})}{2} = \frac{(0.15 + 0.08)}{2} = 0.12$$

thus, for the second row ($H_1$) in Table 1, we have assigned prior probabilities as ($\pi_{1,1}$, $\pi_{1,2}$, $\pi_{1,3}$, $\pi_{1,4}$, $\pi_{1,5}$) = (0.72, 0.44, 0.15, 0.12, 0.08).

Then, for the third row ($H_2$), we have ($\pi_{2,1}$, $\pi_{2,2}$, $\pi_{2,3}$, $\pi_{2,4}$, $\pi_{2,5}$) = 1 − ($\pi_{0,1} + \pi_{1,1}$, $\pi_{0,2} + \pi_{1,2}$, $\pi_{0,3} + \pi_{1,3}$, $\pi_{0,4} + \pi_{1,4}$, $\pi_{0,5} + \pi_{1,5}$) = (0.08, 0.11, 0.15, 0.43, 0.72).

## Supporting information

**S1 Data.**
(R)

## Acknowledgments

The authors really appreciate Dr. Vani Shanker from St. Jude Children's Research Hospital for scientific editing of this manuscript.

## Author Contributions

**Conceptualization:** Chen Li, Haitao Pan.

**Formal analysis:** Chen Li, Haitao Pan.

**Investigation:** Haitao Pan.

**Methodology:** Chen Li, Haitao Pan.

**Software:** Haitao Pan.

**Validation:** Haitao Pan.

**Visualization:** Haitao Pan.

**Writing – original draft:** Haitao Pan.

**Writing – review & editing:** Chen Li, Haitao Pan.

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
