## [Decision Letter · Decision Letter 0]

25 Jun 2020

PONE-D-20-07779

A Phase I Dose-finding Design with Adaptive Shrinking Boundaries and Incorporation of Historical Information -- Extensions of the Bayesian optimal interval (BOIN) design

PLOS ONE

Dear Dr. Pan,

Thank you for submitting your manuscript to PLOS ONE. After careful consideration, we feel that it has merit but does not fully meet PLOS ONE’s publication criteria as it currently stands. Therefore, we invite you to submit a revised version of the manuscript that addresses the points raised during the review process.

Please thoroughly revise your manuscript according the the English language concerns raised by the reviewer.

We look forward to receiving your revised manuscript.

Kind regards,

Jed N. Lampe, Ph.D.

Academic Editor

PLOS ONE

Journal Requirements:

'Pan's research was supported in part by the American Lebanese Syrian Associated Charities

(ALSAC). Yuan's research was supported in part by grants R01 CA154591, P50 CA098258,

and P30 CA016672 from the National Cancer Institute, National Institutes of Health.'

'The funders had no role in study design, data collection and analysis, decision to publish, or preparation of the manuscript.'

6. Please ensure that you refer to Figure 4 in your text as, if accepted, production will need this reference to link the reader to the figure.

Additional Editor Comments (if provided):

Please thoroughly revise your manuscript for English grammar and spelling. It would be useful to have a native English speaker review the manuscript before it is resubmitted.

Reviewers' comments:

Reviewer's Responses to Questions

**Comments to the Author**

1. Is the manuscript technically sound, and do the data support the conclusions?

Reviewer #1: Yes

2. Has the statistical analysis been performed appropriately and rigorously? 

Reviewer #1: Yes

3. Have the authors made all data underlying the findings in their manuscript fully available?

Reviewer #1: Yes

4. Is the manuscript presented in an intelligible fashion and written in standard English?

Reviewer #1: Yes

5. Review Comments to the Author

Reviewer #1: Bayesian optimal interval design (BOIN) is a recently developed model-assisted method for phase I dose-finding trials. This design has been accepted as a popular design in this field due to its simplicity and desirable operating characteristics. In this paper, the authors extend the original BOIN design from two aspects: incorporate the prior knowledge and relax the fixed decision boundaries. Both of these two extensions are important since the 1st one could possibly improve the performance of the phase I trial if relevant prior information can be used and the 2nd is an interesting theoretical exploration of the original BOIN design.

For the 1st extension, the authors proposed an automatic algorithm for its pre-specification of the parameter setups, which makes the proposed method feasibly to be used in real practices. The simulating conclusions are expected, which demonstrates the proposed algorithm works well empirically.

For the 2nd extension, though the results are not so exciting, that is, only when sample size are large, the proposed aBOIN design performs better than the original version, this exploration is still worthy from my perspective since it literally gives a solidified support for robustness of the original BOIN design.

My only concern is that it would be better that the paper can be edited by a professional language editor since there are still grammatic and language inappropriateness.

6. PLOS authors have the option to publish the peer review history of their article (what does this mean?). If published, this will include your full peer review and any attached files.

Reviewer #1: No

---

## [Editor Report · Decision Letter 1]

23 Jul 2020

A Phase I Dose-finding Design with Incorporation of Historical Information and Adaptive Shrinking Boundaries

PONE-D-20-07779R1

Dear Dr. Pan,

We’re pleased to inform you that your manuscript has been judged scientifically suitable for publication and will be formally accepted for publication once it meets all outstanding technical requirements.

Kind regards,

Jed N. Lampe, Ph.D.

Academic Editor

PLOS ONE

---

## [Editor Report · Acceptance letter]

17 Aug 2020

PONE-D-20-07779R1 

A Phase I Dose-finding Design with Incorporation of Historical Information and Adaptive Shrinking Boundaries 

Dear Dr. Pan:

I'm pleased to inform you that your manuscript has been deemed suitable for publication in PLOS ONE. Congratulations! Your manuscript is now with our production department. 

Kind regards, 

on behalf of

Dr. Jed N. Lampe 

Academic Editor

PLOS ONE